# Supporting the emotional needs of young people in care: a qualitative study of foster carer perspectives

Rachel M Hiller ,[1] Sarah L Halligan,[1,2] Richard Meiser-Stedman,[3] Elizabeth Elliott,[1,4] Emily Rutter-Eley[4]

[1]Department of Psychology, University of Bath, Bath, UK
[2]Department of Psychiatry and Mental Health, University of Cape Town, Cape Town, South Africa
[3]University of East Anglia, Norwich, UK
[4]Department of Clinical Psychology, School of Psychology and Clinical Language Science, University of Reading, Reading, UK

**Correspondence to**
Dr Rachel M Hiller;
R.Hiller@bath.ac.uk

## ABSTRACT

**Objectives** Young people who have been removed from their family home and placed in care have often experienced maltreatment and there is well-developed evidence of poor psychological outcomes. Once in care, foster carers often become the adult who provides day-to-day support, yet we know little about how they provide this support or the challenges to and facilitators of promoting better quality carer–child relationships. The aim of this study was to understand how carers support the emotional needs of the young people in their care and their views on barriers and opportunities for support.

**Design and participants** Participants were 21 UK foster carers, recruited from a local authority in England. They were predominantly female (86%), aged 42–65 years old and ranged from those who were relatively new to the profession (<12 months' experience) to those with over 30 years of experience as a carer. We ran three qualitative focus groups to gather in-depth information about their views on supporting their foster children's emotional well-being. Participants also completed short questionnaires about their training experiences and sense of competence.

**Results** Only half of the sample strongly endorsed feeling competent in managing the emotional needs of their foster children. While all had completed extensive training, especially on attachment, diagnosis-specific training for mental health problems (eg, trauma-related distress, depression) was less common. Thematic analysis showed consistent themes around the significant barriers carers faced navigating social care and mental health systems, and mixed views around the best way to support young people, particularly those with complex mental health needs and in relation to reminders of their early experiences.

**Conclusions** Findings have important implications for practice and policy around carer training and support, as well as for how services support the mental health needs of young people in care.

## INTRODUCTION

Young people in out-of-home care are a particularly vulnerable group. The most common reason a child would be removed from their home and placed in the care system is due to the experience of abuse and/or neglect.[1–3] Coupled with maltreatment exposure, children in the care system

have often been exposed to a number of other known risk factors for poor outcomes, ranging from maternal drug and alcohol abuse during pregnancy, to disrupted educational opportunities, and poverty.[4–6] While some young people in care are resilient to their early experiences, it is also the case that poor psychological outcomes are common.[2] A survey of over 1000 children and teens in care in the UK found they were five times more likely to meet criteria for a psychiatric disorder compared with their peers,[7] while a survey of almost 400 seventeen year-olds in US foster care found 61% met diagnostic criteria for at least one psychiatric disorder.[8] Investigating ways to improve mental health outcomes for this group is urgently needed.

**Strengths and limitations of this study**

► This study focuses on an under-researched group (ie, foster carers and young people in their care) and uses qualitative methods to provide in-depth understanding of carers' experiences.
► Young people in care have well-documented high rates of mental health difficulties and while carers are central to providing day-to-day support, we know little about carers' views on their role.
► As this study uses qualitative methods, we cannot conclude whether different styles of support might be associated with different outcomes for young people.
► The study is also focused on carers from a single urban English local authority.

The large majority of children and teens who enter care in the UK will live with a foster carer, who is most often not biologically related to them.[1] These carers are tasked with providing day-to-day support to the young person, including for their emotional well-being.[9] While this relationship can be central for allowing children in care to develop a sense of stability, it can also be complex. To begin with, the foster carer is often a stranger

to the young person, who is entering care following any number of difficult early experiences, including difficult relationships with adults.[10] Placement changes are also common, with one-third of young people in care in the UK experiencing more than one placement in a year.[1] Indeed, it is often those young people with the greatest psychological difficulties who experience more placement breakdowns, with the cycle between children's mental health and placement breakdown being relatively well established, including in research from the UK, USA and similarly developed countries.[10–13]

Qualitative work with young people in care and care leavers has shown that developing trust within foster carer relationships can be challenging. However, factors such as carer patience, consistent boundaries and efforts to spend enjoyable quality time together can help facilitate positive outcomes.[14–17] We know less about supporting young people in care from the perspective of the carers, although there is qualitative evidence of carers' concerns about barriers they face within social care systems, with poor service engagement and support considered detrimental to their role (eg, refs [18] [19]). Beyond this, we know little about how foster carers support the mental health of the young people in their care, despite their central importance to placement stability. The absence of such evidence is potentially particularly problematic, given growing evidence from the broader child mental health field that parents play a key role in facilitating or hindering children's psychological adjustment,[20] including after trauma.[21]

The key objective of the current study was to better understand the role of foster carers, how they support their young people in relation to emotional well-being and what they perceive as barriers and opportunities to providing effective support. Such information has the potential to inform how we might improve the experience of foster carers, and ultimately improve the emotional well-being of the young people in their care. We used a primarily qualitative focus group approach to provide an in-depth investigation of the experience of foster carers, including how they support the mental health of children and teens in their care and navigate the various services involved in decision-making around these young people.

## METHODS
### Sample and procedure
Participants were 21 foster carers, who cared for young people within a moderate-sized urban local authority in England. Sample demographics are presented in table 1, with the sample size based on general guidance for qualitative methodology.[22] While the vast majority of participants were local authority carers, the groups also included private agency foster carers.

We used an opportunity sampling method. Flyers and information sheets advertising three focus groups were circulated via the local authority, foster carer newsletters and social media. Recruitment involved two key methods.

**Table 1** Participant information

| Index | Sample statistics (n=21) |
| --- | --- |
| Age (years), M (SD)* | 51.94 (5.85) |
| Sex of carer, n (% female) | 18 (86) |
| Ethnicity, n (%) | |
| White British | 17 (81) |
| Black British | 4 (19) |
| Additional employment beyond carer role, n (%)* | 8 (38) |
| Living arrangement, n (%) | |
| Partner in home | 9 (43) |
| Biological children in home | 12 (57) |
| Total years as foster carer, M (SD) | 10.39 (8.42) |
| Current number of foster children†, M (SD) | 2.00 (1.14) |
| Sex of foster children currently cared for, n (%)† | |
| Boy(s) | 15 (75) |
| Girl(s) | 12 (60) |
| Age range cared for, n (%)† | |
| Babies | 4 (19) |
| Preschool | 2 (10) |
| School-aged | 16 (76) |
| Adolescence | 11 (52) |
| Type of care provided, n (%)† | |
| Long term | 15 (71) |
| Short term | 13 (62) |
| Respite | 9 (43) |

*Two carers did not report this information.
†Many carers cared for boys and girls, across different age ranges, and across different types of care (ie, long term, short term, respite). Only one carer was not currently caring for any young people.

First, we accessed established support groups within the local authority, and second we used passive recruitment whereby interested carers contacted the researcher for more information on the study. Participants provided informed consent prior to participation. The three focus groups consisted of nine, seven and five participants. All took place in a local community hall and were run by the first author (RMH). Coauthor (EE) also attended each focus group to take notes of broad group and conversation structure to assist with later transcribing. Focus groups ran for between 1 and 2 hours. Focus groups were chosen both based on feedback from carers and services and also to promote dynamic discussions about similar and different experiences. Participants were given the option for follow-up phone interviews if they wanted to express further views more privately, but this was not requested by anyone.

## Patient and public involvement

This study was designed in consultation with carers and service providers (eg, social workers, mental health workers, service managers), who contributed to the objectives and the recommendation to use qualitative focus group procedures to provide a more in-depth exploration of carers' views. Service-users were not involved in recruitment. Dissemination activities for this work have included extensive feedback via written reports and workshops, with multiple services and foster carers.

## Measures
### Descriptive data

Prior to the focus group, all participants completed a self-report background questionnaire, as well as a brief survey on their training and sense of competency for a range of common problems that can effect children in care (eg, based on ref 7). This checklist covered attachment problems, anxiety, depression, post-traumatic stress disorder (PTSD), attention deficit hyperactivity disorder (ADHD), anger and behavioural problems, developmental disorders (like autism spectrum disorder or intellectual disability) and fetal alcohol syndrome. The survey asked the carer to report Yes or No to whether they had received training on (1) 'What the issue looks-like'; (2) 'How best to support the issue'; and (3) 'Whether or not they felt competent in managing the issue'. Carers also answered three questions on 4-point 0 (*Not at all*) to 3 (*Agree a lot*) Likert scales, to provide an index of their confidence across three key questions: (1) *I feel confident that I know the best way to respond to my foster child's emotional needs;* (2) *I feel confident that I know the best way to respond to my foster child's behavioural needs;* (3) *I feel confident that I know the best way to respond to my foster child if they want to talk about their early experiences.*

### Qualitative data

Focus groups were run using a semistructured topic guide that was intended to capture information on (1) the types of challenging behaviours and emotional difficulties that carers have managed, (2) how they cope with, or manage, the emotional and behavioural needs of the children and teens they care for, (3) the positives and negatives of being a carer, and (4) barriers to providing effect support to their foster child.

### Transcribing and analysis

Quantitative information (ie, responses to Likert scales) are presented as descriptives (frequencies). All focus groups were audio recorded and transcribed verbatim. Transcripts were quality checked by another researcher who had not attended the focus groups. Using NVivo software, the transcripts were then coded using a reflexive thematic analysis approach to identify themes and patterns in the data.[23 24] Thematic analysis was used due to the exploratory nature of the research, as it does not require a specific theoretical framework and allowed for a detailed exploration of the data.[23] The coder first read all three transcripts to gain an overview of the data (data immersion), after which each transcript was coded, using an iterative process.[24] Codes were then grouped to form themes, which reflected the core concepts captured across codes. To reflect the complexities of different viewpoints and explore the potential dynamics that may influence carers, each theme was also separated into more fine-grain subthemes. Given the extensive engagement with carers and services in the development of this work, there is potential that the coder, who was also involved in the focus groups, may have been influenced by their own developing views of issues facing carers and services. While the purpose of qualitative research is not to minimise the influence of researcher subjectivity,[24] to explore whether this may have substantially influenced the coding each transcript was read by a second coder who was not involved in the design or data collection, who allocated their own key themes and subthemes blind to the original coding. There was strong agreement between coders, with final codes and themes discussed at a consensus meeting. Key themes were consistent across all three focus groups. There was not capacity within the study to seek further input from participants at this point. However, reflective practices were used throughout all focus groups to ensure clear understanding of the discussion and to minimise any chance of misinterpreting what was being said (eg, reflecting, summarising back to participants and seeking further clarification where needed).

## RESULTS
### Descriptive information

Descriptive statistics are presented in table 1. Foster carers were aged between 42 and 65 years old, were primarily female and had varied levels of experience as carers, ranging from relatively new carers (<1 year of experience) to those who had been foster carers for almost 30 years. Ninety per cent primarily cared for school-aged children and/or adolescents. No carers were biologically related to their foster child(ren).

To understand the training experiences of the participants, foster carers all reported on the types of developmental and psychological issues they had been trained in and on their perception of their own competence. Almost all foster carers reported that they had received extensive training on attachment problems and felt competent managing this, with 68% reporting that they felt competent in supporting anxiety-based difficulties, and 63% reporting feeling competent managing behavioural problems. Far fewer had received training on identifying or supporting other potentially common mental health or developmental difficulties for this group, including ADHD, PTSD, fetal alcohol syndrome and depression (table 2).

From the Likert scale questions, 50% agreed 'A lot' that they knew the best way to respond to their foster child's emotional needs, while a further third (33%) agreed only 'Somewhat' and 17% agreed 'A little'. When asked their

**Table 2**  Percentage of foster carers who had received training on issue and felt competent to manage the issue

| | Had received training on what the issue 'looks like' | Had received training of how to support the issue | Felt competent supporting the issue |
|---|---|---|---|
| Attachment problems | 100% | 80% | 95% |
| Anxiety | 74% | 52% | 68% |
| Behaviour and anger problems | 74% | 58% | 63% |
| Depression | 37% | 42% | 47% |
| Developmental disorders (eg, ASD, intellectual disability) | 47% | 32% | 42% |
| ADHD | 53% | 42% | 37% |
| PTSD | 42% | 32% | 26% |
| Fetal alcohol syndrome | 39% | 17% | 17% |

ADHD, attention deficit hyperactivity disorder; ASD, autism spectrum disorder; PTSD, post-traumatic stress disorder.

level of agreement that they knew the best way to respond to their foster child's behavioural needs 44% agreed 'A lot', 44% agreed 'Somewhat', 6% agreed 'A little' and 6% did not agree at all (corresponding to n=1). Finally, when asked their level of agreement that they felt they knew the best way to respond if their foster child wanted to talk about their early experiences, 68% responded 'A lot', while 16% responded 'Somewhat' and 16% responded that they agreed only 'A little' to this statement.

## Focus group themes

Thematic analysis identified three core themes across all three focus groups: (1) carers were managing very challenging behaviours using parenting intuition and drawing on training; (2) in many, but not all cases, carers were managing these challenges without perceived adequate support from services; and (3) a lack of professional mental health support was particularly problematic, and if it was accessed, perceptions on usefulness were mixed. A breakdown of themes and subthemes is presented in table 3.

### Theme 1: carer strategies for managing challenging behaviour

Participants discussed the types of behaviour exhibited by the young people they cared for, and how they responded to these issues. Responses across all three groups broadly fell into three subthemes.

### Subtheme 1: foster carers often manage extreme and challenging behaviours

For most participants, the most salient examples of their challenges supporting young people were in cases of extremely challenging behaviours that were very difficult to manage. While it was acknowledged that this was certainly not the case for all young people for whom they cared, all focus groups largely focused their discussions on their particularly challenging examples. Participants discussed, at length, the difficult behaviours displayed by some of the young people they currently or previously cared for. Commonly discussed behaviours included aggression, behaviours that were perceived to be manipulative (eg, chronically lying), violent conduct and poor

**Table 3**  Themes and subthemes from thematic analysis

| Theme | Subtheme |
|---|---|
| Carer strategies for managing challenging behaviour | Foster carers often manage extreme and challenging behaviours. |
| | Reliance on training and general parenting techniques to manage challenging behaviours. |
| | It was considered particularly important that the young person had someone to talk to about their experiences, which was usually the carer. |
| Perceived lack of support and adequate training from services | Perceived support from social care and mental health services was often seen as poor and inconsistent. |
| | Perceived support limitations have a negative impact on the young person and the carer. |
| Lack of access to mental health services and mixed views on their helpfulness | Many young people whom foster carers perceived to need mental health support were not able to access it. |
| | Where professional services were accessed views on usefulness were mixed. |

emotion regulation. Managing these behaviours day to day in the home was reportedly extremely challenging.

> Very, very violent to everybody, violent to things, smashing up cars and that sort of thing.

> She did not sleep, she stripped herself naked, she weed all over the place, she was banging herself on the wall.

### Subtheme 2: reliance on training and general parenting techniques to manage challenging behaviours

Participants discussed their application of general parenting techniques and content from training courses in response to challenging behaviours. Responses commonly included practical responses to keeping the young person and broader family physically safe.

> I literally slept on the landing [hallway]. [to keep family safe]

In addition, foster carers described trying to make the child feel psychologically safe, including feeling loved, having the opportunity to share their worries and having stability. In terms of how they learnt these responses, there was little consensus, although many carers suggested they were drawing on their parenting instincts, rather than formal training.

> You can make them your own kids and I think sometimes that is all they want, they just want the normality.

There was also much discussion about '*trying to manage their exposure triggers*', which meant trying to predict situations or factors that might remind a young person of their early experiences and work out how to respond.

> We are all psychologists whether we've had the training or not. You've literally just got to stop for a minute and look at what they're doing and look at what the triggers are and think 'Well what sort of lifestyle did they have? Why is that a trigger?'

This was often challenging as carers were not always aware of the extent of early experiences, particularly when the child was new to them. Views on how to respond to triggers for challenging behaviours varied. When a trigger had been identified, some carers talked about using techniques to support children to face their fears in a controlled and safe way. This was particularly used as a method if they understood where the child's behaviour was stemming from.

> I turned my vacuum cleaner on, s/he'd be hysterical… I said to the [biological] grandma one day about her having [this reaction] and she said 'that would've have been because [grandma explained maltreatment experience related to this reaction]'. So then we worked with the vacuum cleaner, with his/her own vacuum… and I dropped things on the floor ALL the time… Not to traumatise him/her but to say 'it's ok'.

In other cases, identifying the triggers and avoiding them in the future was seen as the best approach, to avoid the child experiencing further distress, as well as outbursts of anger that often accompanied exposure to triggers. For example, in the case of a young person who had particularly difficult memories around kitchens.

> And you can be sitting there and 'oh I'm just cooking come and sit…' [and then you think] 'Oh shit I shouldn't have done that' because all of a sudden pots and pans are flying because it took [the child] into a dark place. So if we're recognising those things and trying to avoid them, really.

Carers also made specific reference to their formal training, with the most commonly discussed training based on the principles of playfulness, acceptance, curiosity and empathy (PACE). Positive and negative aspects of this technique were discussed by participants. All agreed with the principles of this training (eg, the importance of empathy), although participants described difficulties in successfully employing playfulness and humour with their young people who had particularly complex psychological needs.

> Playfulness was very, very difficult.

### Subtheme 3: it was considered particularly important that the young person had someone to talk to about their experiences, which was usually the carer

All carers agreed that to support the child's emotional well-being it was important that they had the opportunity to talk about their precare experiences. All reported that they were often the first person to whom the young person begins to disclose their maltreatment, but some felt ill equipped to manage this or questioned whether it was appropriate for the carer to take on such a therapeutic role without support from services (discussed further later).

> Until the kids start talking, they're not healing.

Of course, young people varied in how open they were when talking about the past, often only disclosing in situations in which they felt safe or more willing to talk, such as when watching TV or in the car. Often carers reported that this might happen in quite unexpected places (eg, while out shopping) so carers needed to be prepared to respond whatever the environment. Participants frequently responded to disclosures by maintaining a safe environment to encourage continued disclosure, such as by putting something benign on TV or extending car journeys.

> One child who used to always talk while the foster carer was driving. And s/he said one day s/he'd just go round and round this roundabout 'cause s/he wanted this kid to carry on talking.

While carers thought these discussions were important for supporting the young person and also for developing

trust and security, they also reported that the training they were provided with in relation to such conversations could be constraining. Caregivers described training as being focused on important concerns with respect to legal aspects of disclosure, such as being careful not to ask leading questions. Many also reported that training suggested that the carer should conceal any emotional response during these conversations. Participants discussed how much of this guidance is difficult to follow in practice and questioned its benefits in terms of the child's emotional well-being. In particular, displaying no emotion was criticised as being an '*impossible*' and potentially damaging response to disclosure.

> [It's] not really giving them [the child] permission to show feeling.

This was especially relevant to the discussion of young people who had demonstrated a limited understanding of their experiences, often expressing confusion, guilt or a lack of emotion and awareness that what happened to them was wrong. Therefore, in what they perceived as a contrast to what they learnt in training, some carers discussed the importance of naming emotions for the young person, so that they can begin to comprehend what happened to them.

> He's got no emotions with it, he's got no feelings, he's got no understanding of it.

> [If they cannot label their emotions] So you're saying 'oh if that was me, I reckon I'd be feeling…'

### Theme 2: perceived lack of support and adequate training from services

Almost all participants reported seeking additional support from services, particularly social care, regarding the mental health of a young person who was, or had been, in their care. However, there was a strong perception that support from services was extremely limited. Most, but certainly not all, carers discussed this as a major barrier in supporting the needs of their young person. Across all three focus groups there were two consistent subthemes.

### Subtheme 1: perceived support from social care and mental health services was often seen as poor and inconsistent

A few participants reported positive relationships with their social workers, and discussed how they were central to supporting the carer and the child, and most recognised that social care systems were under significant resource pressures. Nevertheless, in many cases communication between carers and social workers was described as poor and particularly problematic in terms of being able to effectively support the child. Perceived long delays in the time social workers took to respond to carers played a significant role in this, as they meant that participants frequently were left to manage extremely difficult behaviour unsupported.

> That's what's really hard, it's the waiting time. You're struggling to hold these together, you've got nowhere to turn, or you feel you've got nowhere to turn, you're really managing really traumatised children and you have to wait and wait … and wait.

Participants were also concerned that responses were often inconsistent across social workers. This lack of a clear, universal protocol was perceived as leading to inconsistencies in practice, meaning that the quality of support provided depended on the social worker, rather than the individual needs of the young person. Participants discussed how this inconsistent and often broad-brush approach created challenges for carers in navigating and communicating effectively with social workers.

> If you lined them up and asked them the same question, you'd end up with 40 different answers, and that is scary.

Many participants discussed the negative impact of this poor support as a potential barrier to the relationship between the carer and the child. For example, there was a perception that social services did not always pass along information to carers, or in some cases actively withheld information, particularly in relation to previous behaviour or emotional difficulties. Most participants described how significant information about the young person was often discovered a considerable amount of time into the placement through sources outside of social services, such as previous foster carers or the young person's biological family. Many participants felt that, had this information been passed on earlier, particularly around their maltreatment histories or behaviour difficulties, they would have acted differently to manage behaviour and facilitate their relationship. Some thought this information was withheld as the social worker was worried that the carer would not take the placement if they knew the details of behavioural difficulties.

> Sometimes you find out things six months down the line and you think I wish I had known that at the beginning because you would have done things different. And it, you know, it is very hard. Once that six months have lapsed it's very hard to backtrack.

Overall, communication within and between services (eg, between social care and mental health services), and then with the carers themselves, was seen by most participants as highly problematic and a key hindrance to their ability to advocate for their young people and support their needs.

> None of the systems talk to each other.

> I feel complicit in a system that is not really helping these children it's just housing them, and that feels tragic.

## Subtheme 2: perceived professional support limitations have a negative impact on the young person and carer well-being

The perceived lack of support from outside services was described as affecting both foster carers and the young people in their care. Many foster carers reported feeling exhausted as a result of managing challenging behaviours unsupported. This, in turn, compromised their ability to support the young person with techniques which often require a great deal of energy and consistency.

> We are working 24/7 on very little sleep at times and we are expected to continue, and be playful, and empathetic, and curious and accepting!

Participants also described experiencing 'secondary trauma', relating to the emotional distress experienced in response to young person's disclosures. Participants explained that their training in relation responding to disclosures did not adequately prepare them for how they might feel when hearing the young person describe precare experiences.

> You feel absolutely everything that [the child experienced], and that is horrible.

In light of the limitations to perceived support from services and negative consequences for foster carers' well-being, foster carers perceived the support of their own community to be particularly important. Friendships and online groups within the foster carer community were generally identified as being valuable in providing foster carers a safe space to express their frustrations and support each other emotionally. It also allowed foster carers the opportunity to acquire more practical support, where outside services were lacking, through sharing parenting techniques and advice.

> That's where we get most of our ideas and training.

### Theme 3: lack of access to mental health services and mixed views on helpfulness

Formal mental health services, particularly child and adolescent mental health services (CAMHS; part of the UK National Health Service), were discussed by almost all participants across the three focus groups, who had all had a young person whom they believed required professional support.

## Subtheme 1: many young people whom foster carers perceived to need mental health support were not able to access it

In many cases carers had examples of young people with significant needs who they perceived as being failed by the system because they were not referred for mental health support or they were referred but could not get access.

> S/he could see this black hole… s/he'd hit her head on the wall. I was still left to deal with it all and in the end they sectioned him/her [at 16 years old]. I think from the age of 6 [years old] I was telling them there was something wrong. But you know what we get a lot of is… there's nothing wrong with them, it's

attachment. They love to throw attachment absolutely everywhere.

In many cases where referrals were made, carers discussed extremely long waiting times and increased criteria/thresholds to access treatment. The discrepancy between mental health support accessibility for biological children and for young people in care was also criticised by some participants.

> Now he's in line for CAMHS and by the time he's 21 he'll be there!

> My own daughter suffers from anxiety, I can go and say 'she needs to see somebody', she's seen, she went to CAMHS within 3 months. Yet we've got children that are in the system that have got to wait years.

## Subtheme 2: where professional services were accessed views on usefulness were mixed

Only a minority of cases had successfully accessed mental health support, either via CAMHS, the charity sector or a school counsellor. From those that did, mixed opinions were expressed surrounding the quality of support received. The primary positive of getting a child into professional support was that they were provided with an opportunity to talk about their experiences with a trained professional.

> It's giving him a space, where he can go and um be… to talk really I suppose. He's learning to talk … about things.

> If you get to a place where they'll [child] engage they [mental health professional] will do brilliant stuff.

Key barriers to successful treatment outcomes included the perceived message from CAMHS that the child could not be seen unless they were in a stable placement:

> They can't go to CAMHS until they're in a stable placement. But you can't say that they're going to be in a long term stable placement because you don't know whether or not you're going to be able to look after that child.

Attachment models were often viewed as the blanket response for children in care, without proper assessment of the child's needs. Relatedly, some carers felt their views were not appropriately considered in relation to the psychological support needs of their foster children or teens. There was particularly frustration around the response to requests for support to be the offer of further carer training, without any direct work with the young person:

> … [CAMHS says] it's attachment, it's attachment. I said 'It's not attachment, he is saying some weird things, it's not attachment.

> I don't need for you to tell me how to manage his behaviour, I'm fine, I don't need counselling, I'm alright, it's him that needs the counselling but they

can't do him [see child] until they've done you [further carer training].

[CAMHS says] oh no no, everything's fine…' and we're like 'No no no, I'm with this child 24/7, you have no idea, you have no idea of what that is then…'

And that the young person may not engage with the therapist, meaning sessions were ceased.

The thing with CAMHS they're only any good if the child is willing to engage.

Overall, where young people had accessed support, carers were also keen to be as involved as possible in the therapeutic process. Some reported that they felt left out of the therapeutic process. While they understood considerations around confidentiality, they believed that being more involved, even by just knowing what they should expect from the process in terms of the young person's reaction, would have been helpful in enabling them to appropriately support the young person at home.

I know what they sometimes tell you they don't want us to know and I know it's supposed to be confidential and contained, but sometimes it would be nice for them to give you a little bit of feedback and say 'well in thi- in today's session, they talked about this, this' or 'look out for this, this and this' because sometimes you can have little kids that can come out, I mean I used to get tied to a chair, I used to get things thrown at me, and I used to think 'where's that come from?'.

## DISCUSSION

Foster carers are a primary source of support for young people in care.[9 14] They provide day-to-day support and can be a central advocate for seeking support for the young person within the social care, education and health systems. We examined foster carers' perspectives on supporting the psychological needs of the young people in their care, including how they provide day-to-day support for behaviour or emotional difficulties and their views on the various services involved in the care of these young people. Across three focus groups, consistent themes emerged centred on carers managing extremely challenging behaviours with little perceived consistent and adequate support from social care or mental health services. This was seen to contribute to placement breakdowns, and deteriorations in the well-being of both the young people and the carer.

Given the early experiences of many young people in care and the widely reported statistics on mental health outcomes,[1 7 25] it is perhaps unsurprising that carers reported that they often have to manage a range of significant emotional and behavioural difficulties in the young people they care for. Overall, there was little clear consensus on how best to respond to a young person's needs and only half of the group self-reported that they felt confident managing their foster child's emotional needs, despite all having completed significant training. That said, there were some key areas of agreement. All agreed that it was important for young people to have an opportunity to talk about their past experiences, although many recognised that professional support with this process was important but often lacking. All also agreed that it was important to be empathic and sensitive, although this could be challenging to maintain in the face of ongoing difficult behaviours, and with little perceived support from services.

To support children's mental health some carers described therapeutic-like approaches, providing what might be considered informal exposure techniques to support them to face their fears and worries in a safe space, while some thought trying to avoid triggers that might upset the young person was the best way forward. Confusion around the best way to support children around triggers and fears is potentially problematic, given evidence that avoidant coping is a key maintainer of many psychological difficulties, including for children exposed to maltreatment.[26] The child trauma literature has also shown that parents' own maladaptive response to the child's frightening experiences, such as encouragement of avoidant coping, can lead to worse psychological outcomes for the child.[27–29] However, in the case of foster carers, the young person is typically arriving with already existing emotional and behavioural difficulties, and it remains unclear whether different types of caregiver support could hinder or facilitate psychological adjustment for these young people. It is also worth noting that carers were often reporting trying to manage behaviours and emotional needs that likely required formal psychological support and that this could also impact on their own emotional well-being, which may lead to placement breakdowns.[13] It may be unrealistic to expect carers alone to support the young person to overcome these challenges.

The reliance on their training and accessing support from their own network (especially other foster carers), and often in the absence of professional mental health support, highlights the urgent need to ensure foster carer training programmes are thoroughly evaluated and are evidence based. Our findings reflected findings from mixed-methods research with foster carers in other developed counties, where carers similarly expressed the need for better quality training on supporting the emotional well-being of young people in their care (eg, ref [18]). The importance of ensuring carers are appropriately trained and supported was particularly evident in the current study, where many carers endorsed that they were commonly the first person to whom a young person may begin to disclose the full extent of their early experiences. The literature on disclosures demonstrates that the reactions of adults when children make disclosures of maltreatment can be highly important in influencing both young peoples' willingness to continue to disclose and their mental health.[30 31] Thus, part of training around supporting the mental health and well-being of young

people within their care must also include responding to disclosures in a supportive way. While there has been little systematic evaluation of carer training packages used in the UK care system, a systematic review of foster carer training programmes in the USA found they were often based on little evidence and not thoroughly evaluated,[32] while those which had been evaluated had mixed results in terms of effectiveness.[33 34] Part of the challenge when designing training packages for carers is the lack of evidence for specific foster carer mechanisms that may facilitate children's psychological adjustment. Such quantitative evidence will be key to designing evidence-based training programmes.

Most participants expressed significant frustration with the systems around them and the young person, including social care and mental health systems. There were some positive examples of carers who reported social care had been a central avenue for supporting both their own well-being and the child's well-being. However, it was also the case that almost all had stories of advocating for formal mental health support for their young person, but being unsuccessful either at the social care level (eg, social workers not pursuing the referral) or mental health service level (eg, the young person being deemed ineligible or put on long wait lists), often over many years and despite perceived significant needs. Many had experiences of a young person going on to be homeless, experience teenage pregnancy, self-harming or being sectioned under the Mental Health Act, where early mental health support was not adequately provided. Thus, carers were directly witnessing the consequences of the mental health needs of the young people not being adequately addressed, reflecting widely reported statistics on poor outcomes that continue across the lifespan for those who have been in care.[1 7 25] Consistency of care within social care and mental health services is a long discussed topic.[3] Social work as a profession experiences an extremely high burn-out rate,[35 36] and many high-income countries are facing continued government budget cuts to social care and mental health services. Thus, even with best intentions many services are not adequately resourced to provide support to all young people in need.

Beyond these challenges, findings also showed various avenues for change. Carers recognised that many young people in their care could benefit from psychological support and were often keen to advocate for better support. Thus, the barrier largely does not seem to sit with carers not recognising behavioural challenges as potential reflections of emotional difficulties. If a child could access psychological support, carers were also often keen to be involved, if the clinician thought it would be helpful for the child. Many child-focused interventions involve parent components, which, where appropriate, should also be available for foster carers. Such situations could also provide foster carers with useful diagnosis-specific psychoeducation, which our results showed were less commonly available via their standard training programmes. There is also a clear need for exploring

strategies to improve communication and understanding between foster carers and professionals. Carers were often frustrated at children being labelled with attachment problems, without formal assessments of broader psychological well-being. Further, where mental health services were accessed, the type of support available was inconsistent.

This study has various strengths, including using qualitative methodology to gain a more in-depth understanding of the experiences of carers, and the focus on a group that have received little attention in the empirical literature. Nevertheless, results should be viewed in light of the limitations. First, opportunity sampling means that we potentially captured foster carers with more frustrations with the system, as they volunteered to attend the focus groups to provide their views on supporting young people in their care. Similarly, the semistructured interviews, while largely led by what the participants saw as important to discuss, were focused on supporting children with emotional and behavioural difficulties. While this could capture a wide range of severities of needs and efforts were made to explore positive examples, it was most often the case that the participants chose to focus on children they had cared for who were particularly challenging in terms of their emotional and behavioural difficulties. These are a group worthy of focus, given they are most likely to go through a number of placement breakdowns and have poorer outcomes.[12] However, it is important to note that some children in care do not present with these difficulties and have positive care experiences, or alternatively, do have difficulties that they are able to overcome. Finally, the carers also came from one medium-sized urban local authority, although some had been carers at other local authorities in the past. While results cannot necessarily be generalised, the difficulties within social care and mental health systems across the UK have been well documented and carers' frustrations navigating services also reflect key themes from qualitative work with young people in care in other high-income countries.[18 19]

In sum, results showed that foster carers saw a lack of communication between services, poor support from services and poor access to child and adolescent mental health support as key barriers to them providing effective support to young people in their care, who were struggling with behavioural and emotional difficulties. Where services could not be accessed carers were relying on instinct, support from other carers and drawing on their training. However, many were unsure of the best way to respond to the child's often complex needs and it remains that the evidence base for foster carer training programmes is often scarce, while training or psychoeducation for specific types of mental health difficulties was also uncommon. The availability of foster carers is a central issue for the success of children's services and for supporting vulnerable young people in need. Failing to effectively address the mental health of young people in care does not just impact the child, but can also impact on the mental health and well-being of carers.

**Contributors** RMH is the lead researcher on the project and the PI on the ESRC grant. She led the conceptualisation of the project and project management. She conducted the focus groups and led the write-up of the paper. SLH was involved in the conceptualisation of the project, including the design of the semistructured interview, and contributed to discussions about the themes and feedback on the manuscript. RMS was involved in the conceptualisation of the project and contributed to the discussion about the themes and feedback on the manuscript. EE was the main research assistant on the project and contributed to the focus groups, transcribing, and data analysis, and provided feedback on the manuscript. ERE was involved in the transcription of the focus groups and the data analysis, and was involved in the write-up of the manuscript.

**Funding** This research was funded by an ESRC Future Leader Grant awarded to RMH (ES/N01782X/1).

**Competing interests** None declared.

**Patient consent for publication** Not required.

**Ethics approval** This research received ethical approval from the University of Bath Research Ethics Committee and the participating local authority.

**Provenance and peer review** Not commissioned; externally peer reviewed.

**Data availability statement** Data are available upon reasonable request. Anonymised transcripts will be available on request on a case-by-case basis.

**ORCID iD**
Rachel M Hiller http://orcid.org/0000-0002-4180-8941

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
