## [Reviewer comments · BMJ Open]

ARTICLE DETAILS

TITLE (PROVISIONAL)	Supporting the emotional needs of young people in care: A qualitative study of foster carer perspectives
AUTHORS	Hiller, Rachel; Halligan, Sarah; Meiser-Stedman, Richard; Elliott, Elizabeth; Rutter-Eley, Emily

VERSION 1 – REVIEW

REVIEWER	Hans Grietens University of Groningen, the Netherlands
REVIEW RETURNED	12-Aug-2019

GENERAL COMMENTS	This is a very interesting and well-written manuscript on a relevant topic. The authors address in an in-depth qualitative study the support of foster children's emotional needs by foster carers. The study reveals the complexity of caring for foster children and the negative impact this may have on carers and their families. The study also shows how little support is coming from the foster care system. Another topic that is addressed in this study is the disclosure of early negative experiences of foster kids and the impact of disclosing on foster carers and family systems. Both the content and methods are very clear to me and the overall quality of the manuscript is high. Notwithstanding, I would suggest some revisions in order to further increase the quality of the manuscript and the usefulness of the findings of this study for other researchers and practitioners: - I liked the thematic analysis approach, but I would encourage the authors to go to the latest publications of the developers Braun and Clarke. As they will see, Braun and Clarke reframed their method as 'reflective thematic analysis'. I would like to encourage the authors to stress more the reflective part of the coding process and the identification of the themes. This would further increase the quality of their work.- I would also suggest to have a look at the titles of the themes that are identified. Content wise, the themes are ok, but the titles are quite long. A shorter description to capture the essence of the themes should be possible, no? In addition, the authors should clarify the connection between themes and subthemes, for instance by means of a thematic network (see Braun & Clarke, 2006). Linking the themes (and subthemes) may lead to a better understanding of the complexity of the foster care process, the dynamics and mechanisms involved and the impact of the process on families and family members.- Finally, and just as a suggestion, I would like to refer the authors to the review article by Steenbakkens et al. (2018) on the needs of foster children: Steenbakkens, A., van der Steen, S., & Grietens, H. (2018). The needs of foster children and how to satisfy them: A
---

	systematic review of the literature. Clinical Child and Family Psychology Review, 21(1), 1-12. https://doi.org/10.1007/s10567-017-0246-1. - Since disclosure was identified as a significant theme in the discussions with foster carers, more links to the disclosure literature (focussed on children) could be made in the discussion. - Finally, the authors need to give more arguments for the choice of focus groups as the method of data collection? What is the strength or the added value of this method? Why did the authors not conduct face-to-face interviews with foster carers? Was there a particular reason for choosing focus groups? Did the authors consider face-to-face interviews as well? Both in the methods and the discussion section these questions need to be addressed (in the methods section a clear rationale needs to be given why focus groups were chosen and the discussion section should include a reflection on this choice and the outcome of it).
--	--

REVIEWER	Faith Summersett Williams Ann & Robert H. Lurie Children's Hospital
REVIEW RETURNED	12-Nov-2019

GENERAL COMMENTS	The question of how foster parents support the emotional needs of the young people in their care and their views on barriers and opportunities for intervention is an important and under-studied area. Please do a thorough read through of the paper to correct grammatical errors. Also the introduction can be strengthened by incorporating more background information and citations from studies in the United States and other similar developed countries.
---

VERSION 1 – AUTHOR RESPONSE

Reviewer: 1

Both the content and methods are very clear to me and the overall quality of the manuscript is high. Notwithstanding, I would suggest some revisions in order to further increase the quality of the manuscript and the usefulness of the findings of this study for other researchers and practitioners:

Thank you for your positive and constructive comments on this manuscript.

I liked the thematic analysis approach, but I would encourage the authors to go to the latest publications of the developers Braun and Clarke. As they will see, Braun and Clarke reframed their method as 'reflective thematic analysis'. I would like to encourage the authors to stress more the reflective part of the coding process and the identification of the themes. This would further increase the quality of their work.

We have now updated our language around the analysis, to highlight the reflective process. While being mindful of the word count, we have also added in some brief comments about researcher subjectivity (page 8).

I would also suggest to have a look at the titles of the themes that are identified. Content wise, the themes are ok, but the titles are quite long. A shorter description to capture the essence of the themes should be possible, no?

We have now shortened the theme titles.

In addition, the authors should clarify the connection between themes and subthemes, for instance by means of a thematic network (see Braun & Clarke, 2006). Linking the themes (and subthemes) may lead to a better understanding of the complexity of the foster care process, the dynamics and mechanisms involved and the impact of the process on families and family members.

We have now clarified this on page 8.

Finally, and just as a suggestion, I would like to refer the authors to the review article by Steenbakkers et al. (2018) on the needs of foster children: Steenbakkers, A., van der Steen, S., & Grietens, H. (2018). The needs of foster children and how to satisfy them: A systematic review of the literature. *Clinical Child and Family Psychology Review*, 21(1), 1-12.

Thank you for referring us to this highly relevant review, which we have now referenced in this paper.

Since disclosure was identified as a significant theme in the discussions with foster carers, more links to the disclosure literature (focussed on children) could be made in the discussion.

We have now made further links to this literature (page 21).

Finally, the authors need to give more arguments for the choice of focus groups as the method of data collection? What is the strength or the added value of this method? Why did the authors not conduct face-to-face interviews with foster carers? Was there a particular reason for choosing focus groups? Did the authors consider face-to-face interviews as well? Both in the methods and the discussion section these questions need to be addressed (in the methods section a clear rationale needs to be given why focus groups were chosen and the discussion section should include a reflection on this choice and the outcome of it).

We have now added a clearer comment on our reasons for choosing this method (page 6).

Reviewer: 2

The question of how foster parents support the emotional needs of the young people in their care and their views on barriers and opportunities for intervention is an important and under-studied area.

Please do a thorough read through of the paper to correct grammatical errors. Also the introduction can be strengthened by incorporating more background information and citations from studies in the United States and other similar developed countries.

Thank you for your review of our manuscript. We have carefully proof read the manuscript for grammatical errors. We have also added further references to our introduction and discussion, based on your own and Reviewer 2's feedback. We have also added some more explicit references to similarities across similarly developed countries (e.g., page 4, page 21). Of note, some of the papers we referenced, while published in UK-journals, were reviews of the literature which included articles from across the US, Australia and elsewhere (e.g., Rock et al., 2015).

VERSION 2 – REVIEW

REVIEWER	Hans Grietens KU Leuven, Belgium
REVIEW RETURNED	03-Jan-2020

GENERAL COMMENTS	In this revision, the authors clearly have addressed the reviewers' comments and taken their suggestions. This is a very readable and informing paper with several innovative elements in it, that are adding to the knowledge on foster children's complex behavioural en emotional problems and their needs, as well as their carers' needs. This paper is highly relevant for the practice field. I just have one (small) suggestion: in the Methods section the authors use the word 'patients' (this word is not used elsewhere in the paper). I suggest to replace it by 'children'.
--